# Recent Progress on Graphene Flexible Photodetectors

**DOI:** 10.3390/ma15144820

**Published:** 2022-07-11

**Authors:** Mengzhu Wang, Yingying Xiao, Ye Li, Lu Han, Zhicheng Sun, Liang He, Ruping Liu, Kuan Hu

**Affiliations:** 1Beijing Institute of Graphic Communication, Beijing 102600, China; wangmengzhu777@gmail.com (M.W.); xiaoyingying0420@gmail.com (Y.X.); liye@bigc.edu.cn (Y.L.); hanlu@iccas.ac.cn (L.H.); sunzhicheng@bigc.edu.cn (Z.S.); 2School of Mechanical Engineering, Sichuan University, Chengdu 610065, China; hel20@scu.edu.cn; 3Institute of Materia Medica, Chinese Academy of Medical Sciences, Peking Union Medical College, Beijing 100050, China

**Keywords:** graphene, photodetector, flexible electronics

## Abstract

In recent years, optoelectronics and related industries have developed rapidly. As typical optoelectronics devices, photodetectors (PDs) are widely applied in various fields. The functional materials in traditional PDs exhibit high hardness, and the performance of these rigid detectors is thus greatly reduced upon their stretching or bending. Therefore, the development of new flexible PDs with bendable and foldable functions is of great significance and has much interest in wearable, implantable optoelectronic devices. Graphene with excellent electrical and optical performance constructed on various flexible and rigid substrates has great potential in PDs. In this review, recent research progress on graphene-based flexible PDs is outlined. The research states of graphene conductive films are summarized, focusing on PDs based on single-component graphene and mixed-structure graphene, with a systematic analysis of their optical and mechanical performance, and the techniques for optimizing the PDs are also discussed. Finally, a summary of the current applications of graphene flexible PDs and perspectives is provided, and the remaining challenges are discussed.

## 1. Introduction

With the development of modern communication technology and the Internet of Things (IoT), devices with high foldability, wearability, and bending resistance have been increasingly developed, and there are in-depth studies on flexible optoelectronic devices. Flexible photodetectors (PDs) can be bent, folded, or even stretched, and their applications in imaging, display, optical communications, medical care, and other fields are drawing increasing attention, as they are highly useful in electronic skin, smart textiles, electronic eyes, and flexible cameras [1,2,3,4]. There are usually three components in flexible PDs: flexible substrate, flexible electrode, and functional material. Carbon fiber cloth, fiber, paper, and many polymers, such as polyethylene terephthalate (PET), polyimide (PI), and polydimethylsiloxane (PDMS), are the most commonly used flexible substrates. These widely used substrates for flexible PDs have unique flexibility, high mechanical stability, and high chemical stability [5,6,7]. Indium tin oxide (ITO), as a flexible electrode, is widely used in displays. However, due to its inherent fragility, alternatives to this material have been developed, such as conductive polymers and metal nanowires, among which transparent conductive graphene films have been widely studied in recent years. The functional components in flexible PDs include materials such as two-dimensional (2D) perovskites and 2D metal halides [8,9]. Flexible PDs should maintain stable performance during repeated bending, folding, or stretching [10], placing a high demand on the mechanical stability and flexibility of their component electrodes and functional materials. Due to their adjustable bandgap, high light absorption efficiency, electron mobility, and low sensitivity to their environment, carbon materials have great potential in the field of PDs [11]. Graphene was the first known 2D layered material, with its discovery leading to a research boom due to its high flexibility, extraordinary elastic modulus, and large strain (>10%). Furthermore, optically transparent graphene exhibits high carrier mobility, and its optical performance can be adjusted via electrostatic doping or strain to realize flexible electronic devices with novel functions. In addition, the high carrier mobility of graphene ensures the ultrafast conversion of photons or plasmons into electrical signals, which is very helpful for producing ultrasensitive PDs exhibiting high photoconductivity gain [12,13]. In graphene-based PDs, to achieve effective light capture, it is necessary to increase the light absorption rate, establish a suitable band structure, and improve the quality of the interface. When graphene-based PDs were first studied, the main focus was on analyzing the structures of field-effect transistors. Early studies showed that the light response rate of these PDs was generally < 0.01 A/W [14,15]. In recent years, many studies have been carried out to improve the performance of graphene PDs, with many studies focused on the preparation of graphene conductive films. As one of the most important parts of a PD, the performance of graphene determines the device’s performance.

Another important strategy is to develop graphene hybrid structures. Graphene can be used as a light-absorbing layer or a transparent conductive layer in devices [9]. It can be doped or heterocomposited with many semiconductors (SCs), quantum dots (QDs), transition metal dihalides (TMDs), and perovskites that exhibit excellent photoelectric performance. The spectral range, photodetection efficiency, and responsivity of the resultant hybrid PDs greatly exceed those of materials based on a single component [16,17]. Herein, the current research on graphene-based conductive films for photodetection is thoroughly analyzed, and the latest developments in this area are summarized, especially those relating to chemical vapor deposition (CVD) technology. In the Section 2, the development of all-graphene photoelectric detectors is briefly introduced. The Section 3 focuses on the research of high-performance graphene hybrid PDs, in particular the latest developments in new materials and device design. According to several important parameters of flexible PDs (external quantum efficiency (EQE), detectivity, response rate, response time, and photoconductive gain) that are used to effectively evaluate devices, various methods for enhancing the performance of graphene PDs are discussed, with analysis and comparisons. In the Section 4, the future applications of PDs in flexible wearable electronics are discussed. In the Section 5 an overall summary and perspectives for the development in this field is provided.

## 2. Two-Dimensional Graphene

Carbon-based materials, such as diamond, graphite, and amorphous carbon, have broad applications. New carbon materials, including zero-dimensional (0D) fullerenes, one-dimensional (1D) carbon nanotubes (CNTs), and 2D graphene, have gradually evolved into the most promising carbon nanomaterials. Graphene is a single-layer sheet of graphite, which is the matrix of other allotropes of carbon, such as fullerenes, CNTs, and graphite [18,19,20,21]. It is a stable 2D atomic crystalline material formed from covalently bonded carbon atoms, arranged in a honeycomb lattice, and has a unique Dirac tapered energy band structure. The thickness of this monoatomic layer of graphite is approximately 0.335 nm [22,23]. In 2004, at the University of Manchester, Novoselov and Geim used a tape stripping method to strip graphene from graphite crystals and prepare graphene devices, and there has been exponential progress in the study of graphene for applications in various electronic devices [24].

Due to its 2D structure, graphene has unique physical properties, such as high electrical conductivity and high light transmission. Ultrathin graphene also exhibits high chemical stability, high ductility, and high mechanical strength, e.g., monolayer graphene has a high Young’s modulus of 1.0 TPa and a tensile strength of 130 GPa [22]. Graphene was the first 2D material prepared to exist stably at room temperature. These excellent characteristics make it widely applied in photoelectric detection. Since graphene is a semimetallic material with no gap, it has an ultrahigh carrier mobility of as high as 20,000 cm^2^/(V·s) at low temperatures [24], a value that is 10 times greater than those of silicon materials and twice that of indium antimonide (InSb), which has the highest known carrier mobility, as the electron mobility of graphene is less affected by temperature. At any temperature in the range of 50–500 K, the electron mobility of monolayer graphene is approximately 15,000 cm^2^/(V·s) [25,26]. Graphene can maximize the gain of PD devices and exhibits a wide response range across the entire electromagnetic spectrum. Due to its zero bandgap, it can absorb light over a wide spectrum (from the visible to terahertz (THz) regions) [22,23], providing the possibility of light detection over a wide spectral range. To the best of our knowledge, it is the only known conductive material that exhibits high transparency across the entirety of the infrared (IR) region of the spectrum (including the mid-to-far IR), as shown in Figure 1. The excellent ductility of graphene also expands its application prospects in flexible optoelectronic devices, and it is expected to have broad development prospects in the fields of electronics, photonics, energy, environmental protection, and biomedicine.

### 2.1. Preparation of Graphene Transparent Conductive Films

At present, the crystal domain size of graphene films is mostly from micrometers to millimeters, and it can even reach the centimeter level [27]. The current trend in the development of graphene films is to develop the controllable, rapid preparation of graphene films with a large area and large crystal domains via high-quality in situ deposition [28]. Due to the interlayer aggregation of graphene and its small active area (~100 μm^2^), it is difficult to realize uniform deposition on the required substrate. Therefore, the efficiency of graphene-based (Transparent Conductive Films) TCFs is generally lower than that of commercially available transparent conductive oxides. Graphene films used in photoelectric detection not only need to be highly efficient but must also be uniform without agglomeration [29]. Recently, the Li research group proposed a new solution to eliminate the problem of graphene wrinkles. The study found that a high proportion of hot hydrogen can overcome the force between graphene and the substrate to a certain extent, with the protons and electrons in the hydrogen passing through the graphene layer, which makes the prepared multilayer (ML) graphene exhibit better layering, with an almost wrinkle-free appearance [30].

Although the development prospects of graphene films are great, the large-area and high-quality preparation of graphene films remains a critical challenge. Graphene was originally obtained via the mechanical exfoliation of graphite flakes. At present, a series of preparation methods for graphene have been developed, including liquid-phase exfoliation, reduction of graphene oxide, epitaxial growth of silicon carbide (SiC) or metal single crystals, molecular assembly, and CVD, among others [24,31]. The graphene prepared via mechanical exfoliation is of high quality, but its small size and low yield limit its wide application. A large amount of graphene can be prepared by chemical reduction, but its electrical performance is relatively low. Wafer-thin graphene can be prepared through SiC epitaxy, but it does not meet the requirements of large-area flexible electronic devices, and it is difficult to transfer the graphene attached to the SiC substrate to a flexible substrate. Molecular assembly is an expensive method, meaning that graphene prepared in this way does not meet the requirements of low-cost flexible electronic devices [32].

These methods are thus not suitable for application in flexible electronic devices. CVD is another effective technology for synthesizing large-area graphene films. At present, this technology is being widely studied in the field of graphene film preparation [33]. In the CVD process, gaseous or vaporous substances react in the gas phase or on a gas–solid interface to generate solid deposits. The preparation of graphene films via CVD usually requires a high-temperature furnace. Under high-temperature conditions of approximately 1000 °C, carbon-containing precursors such as methane are used as carbon sources to pass into the high-temperature chamber, and processes such as carbon source cracking, surface diffusion of activated carbon species, and graphene nucleation growth occur on the substrate, finally leading to a graphene film being deposited on the substrate [29].

Thermal CVD of graphene involves the use of first-row transition metals, such as iron (Fe) [34], cobalt (Co) [35], nickel (Ni) [36], and copper (Cu) [37,38] as catalysts. The solubility of carbon in these metals is the main parameter that affects growth quality. Among them, Fe has the highest solubility and Cu has the lowest solubility. Therefore, Cu is the preferred catalytic metal for the growth of monolayer graphene. In 2009, the Ruoff group successfully prepared a large-area graphene film on Cu foil (25 μm) for the first time via CVD. This process can be used to grow graphene on a 300 mm copper film on a silicon substrate, which is a very important technological breakthrough [31]. The film has the advantages of high quality and good controllability. Since then, the road to large-scale preparation of high-quality graphene films has opened up. For a long time, CVD was considered to be the most promising method for preparing large-area high-quality graphene films. However, as the size of the reactor increases, there are significant increases in manufacturing difficulties and the cost of CVD reactors, which in turn lead to limitations on the size and throughput of graphene films [39]. In addition, the high-temperature CVD growth of graphene is accompanied by some side reactions, leading to a large amount of amorphous carbon contaminants being deposited on the surface of graphene, leading to “intrinsic pollution” of the graphene film, which seriously affects the performance of graphene. Currently, there is still a large discrepancy between the measured performance of graphene films and the values expected from theory, indicating that research is required to improve the performance of graphene films [40,41]. Wang et al. [39] reported a “breathing” CVD method, in which a spiral Cu foil as substrate is employed to increase the loading density. In this process, graphite spacers are placed between the Cu layers at both ends of the spiral to prevent the Cu layers from adhering together at high temperature. There is enough space inside the spiral substrate, and the reaction gas is inhaled by cyclically adjusting the rise and fall of the pressure in the reactor and the Cu spiral of the auxiliary gas exchange. This method is similar to breathing, effectively using the space of a small reactor, and the size and throughput of the prepared graphene film is an order of magnitude higher than that of traditional methods. To date, the temperature required in most reported CVD methods is above 1000 °C to achieve the complete preparation of graphene. Aside from it being a time-consuming and high-cost method, the most important point about CVD is that it is not suitable for basic materials that are not resistant to high temperature. Recently, a new CVD method has been developed that is different from traditional processes. The new method utilizes molten gallium as a catalyst and sapphire and polycarbonate as substrates. Using this method, graphene can be grown at approximately 50 °C. This technology can therefore overcome the high-temperature requirement of traditional preparation methods [42,43].

### 2.2. Transfer Technology of Transparent Conductive Graphene Films

The preparation of high-quality graphene is an important issue in the practical applications of graphene. Simultaneously, transfer of graphene is also an indispensable process, which is closely related to the realization of large-scale production of graphene films. For application of graphene in PDs, it needs to be transferred to a target substrate that is suitable for the device. As mentioned in Section 2.1, due to the limitations of technology and preparation conditions, it is difficult to obtain clean graphene via a direct growth method, and the transfer process makes the film even dirtier. Many impurities are deposited on the surface of the film, which have an adverse effect on the performance of the device. Therefore, the surface contamination of graphene is a critical and unresolved challenge [41]. In addition, the transfer process of graphene is also complicated, as graphene is prone to the formation of wrinkles, folds, or defects during its transfer process, all of which reduce its transfer efficiency. The transfer of graphene can be divided into two categories: direct transfer and indirect transfer.

### 2.3. Indirect Transfer Method

Using a carrier material as a supporting layer, after the graphene is transferred from the growth substrate to the target substrate, the carrier material is removed through physical or chemical methods to complete the transfer. The common method is using polymethyl methacrylate (PMMA) as the carrier to obtain the transfer of graphene grown on the surface of Cu foil. PMMA has high flexibility and high solubility in a variety of organic solvents, but its most important property is high transparency, which is beneficial for clearly observing the removal process of the Cu foil [44]. However, this method will easily lead to damage and wrinkling of the graphene. PDMS can also be used for transfer [26]. A benefit of using PDMS is its lower surface energy compared to that of PMMA, making it easier to separate the polymer from graphene after its transfer. Zhang et al. [45] selected slow-adsorption and low-cost rosin as a support layer to complete the clean and structure-intact transfer of large-sized graphene and successfully produced a single-chip flexible organic light-emitting diode (OLED). Leong et al. [46] took advantage of the stable chemical properties of paraffin and non-covalent adsorption, replacing PMMA with paraffin to achieve high-quality transfer of graphene. Through a combination of the roll-to-roll technique and other automated processes, the temperature during transfer can be well controlled, and the efficiency and production yield of transfer can be greatly improved. Additionally, through reasonable selection of the carrier and continuous optimization of the process, indirect transfer can be carried out to obtain high-quality graphene. In theory, this method can be used to transfer graphene to any substrate and has a wide range of applications.

### 2.4. Direct Transfer Method

Using this method, graphene is directly attached to the target substrate and peeled off from the growth substrate to complete the transfer without the need for any carrier materials. The principles behind this method are that after the target substrate is processed, the binding force between graphene and the target substrate is much larger than the binding force between graphene and the growth substrate, with the growth substrate then being removed via etching or direct peeling. In 2012, Yoon et al. [47] used epoxy resin (EpoTek 353ND) to directly peel off graphene from a Cu substrate and used a double cantilever beam (DCB)-based fracture mechanics test to directly measure the adhesion between graphene and the Cu substrate for the first time, which proved that the direct transfer of graphene from the Cu substrate to the target substrate could be achieved. Compared with the indirect transfer method, the direct transfer method has no need to remove the carrier due to the addition of an assistant interlayer, and cracks will not be introduced in the graphene due to its spontaneous relaxation issues during the degumming process, greatly reducing the damage rate of the graphene film, and thus making it stronger. However, attention needs to be paid to the issue that the adhesion interlayer cannot have a great negative effect on the performance of graphene. Considering that the interlayer is inevitably a source of pollution, studies are also focused on attempts to directly transfer graphene to the target substrate without the assistance of a polymer. For example, Lin et al. [48] proposed adding isopropanol to adjust the surface tension of etchants to protect graphene from tearing, thereby realizing direct carrier-free transfer. However, most of these methods require strict experimental conditions, which are limited by the film size, equipment, substrate, and other conditions; therefore, production efficiency and cost cannot be balanced.

For the transfer of graphene, the solution processing of graphene offers various facile processes, such as spin, dip, and spray coating. The production cost of this method is relatively low and involves the use of fewer chemicals. However, overall, the high-quality, uniform, easy-to-process, and low-cost transfer of graphene is still challenging and requires further exploration to meet industrial and market demands [49].

## 3. Full Graphene PDs

The basic principles of traditional PDs are straightforward and are described as follows. A reverse bias is applied to PN or PIN junctions; when incident light with a higher energy than the bandgap of the SC absorption layer irradiates the depletion region, the light is absorbed, and photogenerated carriers are generated. The photogenerated electron–hole pairs are separated under the action of an external electric field and quickly reach the electrode on both sides, thereby generating a photocurrent to detect optical signals [50].

In 2009, Xia et al. [51] developed the first graphene PD using graphene as a light-absorbing layer. They achieved this PD by adjusting the back gate bias voltage to control the barrier height between the metal and graphene to change the channel carrier concentration and obtain ultrafast photoelectric detection, mainly to test the changes in the device in the dark and in light. When analyzing and calculating the test results, it was found that the bandwidth of the graphene detector can reach 40 GHz without attenuation and, according to their predictions, the working bandwidth of the graphene PD can be further increased to 500 GHz. Subsequently, Mueller and coworkers [52] used an asymmetric interdigital electrode to obtain graphene PDs with improved performance. Due to the interdigital electrode, the built-in electric field is enhanced, and the photocurrent increases. At a wavelength of 1.55 μm, the photo response of the detector reaches 6.1 mA/W, realizing optical data link communication at 1550 nm of 10 Gbit/s. However, the existence of the metal shielding effect limits the photoresponsivity of the device. In 2012, Dong et al. [53] exploited the ultrahigh electron mobility of graphene to produce a fast time-response PN junction detector. Using an ultrafast pump probe, they measured the photocurrent response time of the detector as approximately 1.5 ps at room temperature and found that this response time increased to 4 ps at 20 K. With the development of CVD technology, Zheng et al. [54] demonstrated a low-temperature metal-free plasma-enhanced CVD method to grow large-scale graphene to fabricate an all-graphene PD. The detector has a high response rate in the visible range, with the maximum photogenerated current reaching the mA level when the wavelength is 405 nm. The response rate is as high as 250 A/W, the detectivity is 1.3 × 10^11^, and the maximum photoconductivity gain is 1.6 × 10^11^. These results are orders of magnitude higher than previously reported results (response rate of 6.4 A/W, detection rate of 6 × 10^7^, and photoconductive gain of 3.7 × 10^4^) [55].

However, graphene alone does not have a bandgap, since its unique structure causes photogenerated electron–hole pairs to recombine too fast, meaning that they cannot be effectively separated, limiting its light absorption and EQE. This zero bandgap will also cause the dark current of the graphene device to be too high, indicating that graphene PDs do not achieve a high degree of detection. In addition, although graphene absorbs over the entire electromagnetic spectrum, due to its extremely low light absorption coefficient, intrinsic graphene can only absorb 2.3% of incident light for graphene with a single-layer thickness of 0.34 nm [56]. These weak light absorption characteristics limit the concentration of photogenerated carriers, showing that the response and sensitivity of detectors featuring graphene as the active layer are very low [51,52]. Based on these factors, when the intensity of incident light is weak, the feasibility of using a graphene detector is a problem, which is also a critical factor that limits its applications. Therefore, when graphene is applied in PDs, stronger light absorption is needed. For photoresponsiveness, in hybrid graphene/QD photodetectors, photogenerated charges can be transferred from colloidal quantum dots to graphene, and carriers with opposite charges are still trapped in the QD layer. After the holes reach the drain, due to the charge conservation in the graphene channel, the source will supplement the holes, and multiple holes circulate in the graphene channel, resulting in the optical gate effect. The most successful method to enhance the light response is the optical gate effect of graphene. When using QDs, the photoresponsivity can be as high as 10^7^ A/W, but it will directly affect the response time, and the overall effect is not ideal [57]. Theoretically, the responsivity of the device increases in line with an increase in the number of graphene layers; for example, the responsivity of a PD with 20 layers of graphene is 74% [58]. However, as the number of layers increases, the fabrication process becomes very complex, resulting in low production of graphene. Over the past few years, various attempts have been made to improve the performance of graphene PDs. Among them, plasma enhancement [59], resonant cavity enhancement [60], and heterostructure recombination [61] are widely explored [62]. On the other hand, for photodetectors that need to detect specific wavelengths, other wavelengths can be filtered out in different ways. For example, Thompson et al. [63] used hexagonal boron nitride (h-BN) as an absorbing medium for UV photodetectors because its band gap was wide enough to be insensitive to light in the range where it can be detected. Testing has shown that the device is photoresponsive to a 254 nm light source, but not to a 365 nm source.

As we know, the most important parameter for evaluating a PD is the specific detectivity, *D**, which reflects the sensitivity of the PD. The realization of high *D** relies on a high light response rate and very low noise [64]. In 2013, Gan et al. [65] used metal-doped graphene to fabricate an integrated waveguide PD, which simultaneously exhibits high responsiveness, high speed, and wide spectral bandwidth. The device responds almost uniformly under illumination from 1450 to 1590 nm, with a light responsivity of >0.1 A/W. Wang et al. [66] coupled graphene with silicon waveguides to achieve broadband optical detection at room temperature. By inserting graphene into an atomically thin PN junction, a MoS_2_-graphene-WSe_2_ heterostructure for wide-band optical detection in the visible short-wavelength IR region of the electromagnetic spectrum at room temperature was prepared. The bandwidth of this device reached 400–2400 nm, and the *D** of the heterojunction in the near-IR region reached 10^11^ Jones [67] Niu et al. [68] used CMOS nanotechnology to fabricate a single-layer graphene/Ge/Si-tip Schottky junction structured PD. The silicon substrate was a silicon tip with a width of 50 nm, and then a highly selective germanium (Ge) nanoisland was grown. There were no extended defects in the Ge islands, so the photoelectric detection characteristics were high, and the responsivity and switching ratio were determined to be 45 mA/W and 10^3^, respectively. Such hybrid PDs that have undergone doping or heterogeneous recombination exhibit greatly improved light responsivity and *D** values. The following section gives an introduction to graphene hybrid PDs, including their chemical doping and heterostructure composites.

## 4. Graphene Hybrid PDs

The room for the development of hybrid PDs is huge. The resistivity of this type of detector is low, and the carrier mobility is extremely high. The response rate and speed of the device are also high, which is exactly what is required for PDs. This is also the reason why the performance of graphene hybrid PDs is higher than those of traditional (semiconductor photodetectors) SCPDs [54].

### 4.1. All-Carbon (Graphene–Carbon Allotrope) PDs

As a type of low-dimensional carbon material, SC C_60_ exhibits significant absorption from the ultraviolet (UV) to near-IR regions. Its medium bandgap allows the suppression of dark current and low noise photoelectric detection. Due to its excellent optoelectronic properties and mechanical properties, it has a wide range of potential applications in low-cost flexible optoelectronic devices. The combination of C_60_ and graphene, a material with low light absorption, overcomes the defects of all-graphene PDs, and the unique semiconductor characteristics of C_60_ also promote the performance enhancement of all-carbon flexible PDs. Qin et al. [69] obtained a well-crystallized C_60_ film on a hexagonal boron nitride template. Two graphene sheets were used as the source and drain electrodes to prepare a graphene-C_60_-graphene PD operating over 200–800 nm. As an all-carbon device, it has an operating bandwidth of approximately 5 kHz and an extremely high responsivity of 5510 A/W at 360 nm and 2280 A/W at 405 nm. The graphene-C_60_ interface barrier with a tunable gate significantly affects the responsivity of the device, with the principles behind this shown in Figure 2a. Then, they demonstrated an all-carbon PD based on a van der Waals graphene-C_60_ heterostructure on a plastic substrate. Figure 2b,c shows the manufacture of the graphene-C_60_ device on a PET substrate. It is considered the first all-carbon flexible PD fabricated using a graphene-C_60_ hybrid structure that exhibits long-term stability. The tight electronic coupling at the all-carbon interface in such devices leads to improved charge transfer efficiency. Due to the excellent light absorption capability of C_60_, the resultant detector can be used to detect bright signals, including fluorescent lamps, and the light response in the UV to near-IR regions is relatively significant. The mechanical flexibility of the detector is also very high, as shown in Figure 2d–f. After hundreds of bending cycles under extremely high tensile strain, there is only a very slight change in the light response of the device [70].

As one of the allotropes of carbon, 1D CNTs are rolled from a single-layer graphene sheet. They are hollow cylinders with a diameter of approximately 0.5–2 nm that are very flexible in the radial direction and have great potential in high-performance nanoelectronics [72]. The mobility of single-walled CNTs (SWCNTs) can reach approximately 40 cm^2^/(V·s), and their on/off ratio is approximately 10^5^ [67]. These characteristics make CNTs highly suitable as semiconductor channels. Experimental results showed that the integration of a CNT network with polycrystalline graphene obtains enhanced mechanical strength and improved conductivity of the resultant device but does not have any impact on optical transparency [73,74,75,76]. Thin-film transistors based on SWCNT-graphene heterostructures can operate under a strain of ≈20% while still exhibiting high mechanical stability and extremely high transparency [77]. Liu et al. [78] reported, for the first time, a graphene SWCNT flexible PD, which exhibits a high photoresponse of ≈51 A/W and a fast response of ≈40 ms in the visible range, as well as high robustness against repetitive bending. Pyo et al. [71] modified a SWCNT network with porphyrin and demonstrated a heterogeneously integrated SWCNT-graphene PD. As shown in Figure 2g–j, the device exhibits a high photoresponse (1.6 × 10^−2^ A/W), excellent mechanical performance, and stable performance after heavy crumpling (≈50% strain). The experimental findings show the great potential of this device as a wearable sunlight sensor that can be used on human skin and other curved surfaces.

All-carbon hybrid films are highly robust films. Their hybrid structures are fully compatible with graphene-based electronic components and can facilitate the integration of large-scale optoelectronics. The hybridization of low-dimensional carbon nanomaterials also allows the realization of extremely flexible, stretchable, and transparent electronic devices. Moreover, the relatively low cost of flexible all-carbon nanomaterials means that they have considerable potential for applications in future wearable and foldable PDs. However, the high optical transmittance of such all-carbon devices has a negative effect on their light response across the visible spectrum, demonstrating that effective methods are required to increase the light-absorption rate while ensuring the transparency of these devices. These methods include decorating or integrating microcavities and waveguides with nanoparticles (NPs) or graphene/SWNTs to improve the photoresponse of carbon nanomaterials [66,79,80]. For example, Muench et al. [81] proposed a waveguide integrated plasma-enhanced graphene PD exhibiting a 3 dB band width of ~42 GHz. This is the highest value for a PET-based on-chip graphene photodetector made from CVD SLG (single-layer graphene). The photoresponsivity of the device has also been greatly improved, and the external responsivity is ~12.2 V/W.

### 4.2. PDs Based on Chemically Doped Graphene

Topological insulators that have a conductive surface state and exhibit overall insulation properties have been widely investigated for various applications. A large number of studies have shown that topological insulators exhibit excellent photoelectric, high light absorption, and strong photoelectric conversion properties [82,83,84]. Bismuth telluride (Bi_2_Te_3_), as a typical topological insulator, is similar to graphene, exhibiting hexagonal symmetry, a narrow bandgap, high carrier mobility (10^4^ cm^2^ V^−1^ s^−1^), high stability, and great prospects in PDs [85,86]. Qiao et al. [87] used CVD to epitaxially grow Bi_2_Te_3_ nanocrystals on single-layer graphene and fabricated a PD based on this graphene/Bi_2_Te_3_ heterostructure. Figure 3a shows a band diagram of this heterojunction. At the interface between Bi_2_Te_3_ and graphene, a light-carrying current is effectively generated and transferred without sacrificing the width of the detection spectrum to effectively increase the photocurrent of the device. It is found that the photoresponsivity and sensitivity of the graphene/Bi_2_Te_3_ PD are higher than those of the pure graphene-based PD (35 A/W at a wavelength of 532 nm, photoconductive gain up to 83), and the photocurrent of the heterostructure-based PD is approximately 10 times that of the pure graphene-based PD without applying a grid bias [88]. Experimental results showed that the detection wavelength range of the device can be further extended to the near-IR (980 nm) and telecommunication (1550 nm) bands, which is not observed for devices based on graphene and transition metal dihalide/carbon heterostructures. Unfortunately, this PD is not self-powered and needs to be driven by external energy, and the process of manufacturing Bi_2_Te_3_/graphene heterostructures by CVD is relatively complicated, suggesting that these materials are not suitable for large-scale preparation. Some reported results demonstrated that this heterostructure-based PD can be manufactured by a facile one-pot hydrothermal process, with the prepared composite material as an electrode material in a photoelectrochemical (PEC) PD. As described in the photoelectrochemical results, the prepared Bi_2_Te_3_/graphene heterostructure-based PD exhibited photoresponse performance when exposed to sunlight. Moreover, as the PEC PD is self-powered, it can operate under a bias potential of 0 V, and its light response is very obvious, and can reach 2.2 mA/W. At the same time, upon an increase in the irradiation power density, there is a linear increase in the photocurrent density of Bi_2_Te_3_/graphene heterostructure-based PD. Even when the light intensity is reduced, the photocurrent density still reaches a high level, proving that the Bi_2_Te_3_/graphene heterostructure-based PD still shows strong light detection ability under low light intensity [89]. In addition to Bi_2_Te_3_, Yoo et al. [17] recently used Bi_2_Te_3_ nanowires (NWs) to modify graphene-based PDs that delivered 3 × 10^4^ high gain and a wide bandwidth window of 400–2200 nm. Compared with the PD using a Bi_2_Te_3_/graphene nanoplate junction, the photoconductive gain of the device can be improved by two orders of magnitude. Using a low bandgap Bi_2_Te_3_ NW and graphene junction, the light response rate at 2200 nm (~0.09 mA/W) can be increased by 200%. These Bi_2_Te_3_ NW/graphene heterostructures have the advantages of high flexibility, low cost, and facile preparation, and are highly suitable for applications in wearable pressure and light sensors. There are many optoelectronic materials with this topology, such as 2D covalent organic frameworks (COFs), which can also be combined with graphene to form heterostructures. COFs have a high specific surface area and positive selectivity, which enables the photosensitive characteristics of PD to be reversibly adjusted using specific target molecules; therefore, the performance of the device is improved [90]. 

As a transparent conductive semiconductor, ZnO has a wide bandgap of ≈3.37 eV and a high exciton binding energy of 60 meV [91]. The novel electrical and optical properties of 2D nanostructured ZnO provide it with great prospects in optoelectronics, such as in graphene/ZnO heterojunction flexible PDs, which can detect broadband light in the UV-to-visible range at room temperature, compared with a single-component PD, with the photoconductivity of the device operated under an external voltage being greatly improved [92,93]. Recently, ZnO nanorods (NRs) have been reported to exhibit excellent properties, such as a large specific surface area, a high density of trap states that increase the carrier lifetime, low charge transfer resistance and higher electron–hole separation [91,94,95]. In addition, the work function of ZnO is very close to that of graphene. Based on this, Dang et al. [96] developed a hybrid ZnO NR and graphene phototransistor. This flexible transistor device showed a maximum light response at 365 nm, with the highest response rate of 2.5 × 10^6^ A/W and a photoconductivity gain of 8.3 × 10^6^. These high values can be attributed to the ZnO nanorods and 5 V grid modulation. This 5 V grid modulation allows the quasi-Fermi level of ZnO to approach the conduction band of graphene, reducing the energy barrier between the materials and enhancing the transfer of photogenerated electrons from ZnO NRs to graphene [97]. This structure was tested under 0.5% tensile strain for 10,000 bending cycles (curvature radius of 12 mm), and the response rate was found to remain the same. Under the action of mechanical stress, the flexible PD based on ZnO NR/graphene exhibited high stability and high interface protection. The introduction of a semiconductor can effectively improve the sensitivity of the device. Graphene produced via CVD can be transferred to a flexible PET substrate. Wang et al. [98] produced a flexible PD based on a graphene/ZnO NW array, a schematic diagram of which is shown in Figure 3b. The piezoelectric polarization charge at both ends of ZnO NWs is used as a charged impurity, which effectively disperses the transmission of electrons in the graphene. The flexible substrate has an excellent response under bending strain, which indicates that compressive and tensile strain have the same effect on the current change, as shown in Figure 3c. The light response rate of this device at a wavelength of 325 nm is 3.5 µA/W; additionally, it has high photosensitivity and a response speed of 28/25 ms. Compared with traditional optoelectronic devices, flexible devices usually have poor performance due to interfacial problems. The interfaces of optoelectronic devices can be optimized by improving the contact between their layers. The discontinuity between the functional layer and substrate also reduces the performance of the device. Liu et al. [99] introduced perovskite nanosheets between ZnO and graphene layers and found that the photomultiplier tube showed an illumination ratio of ~10^3^, an order of magnitude superior to that of a photomultiplier without CsPbBr_3_. The performance of this flexible PD is stable, with its photocurrent exceeding 93% of the maximum value after being bent 500 times [100].

**Figure 3 materials-15-04820-f003:**
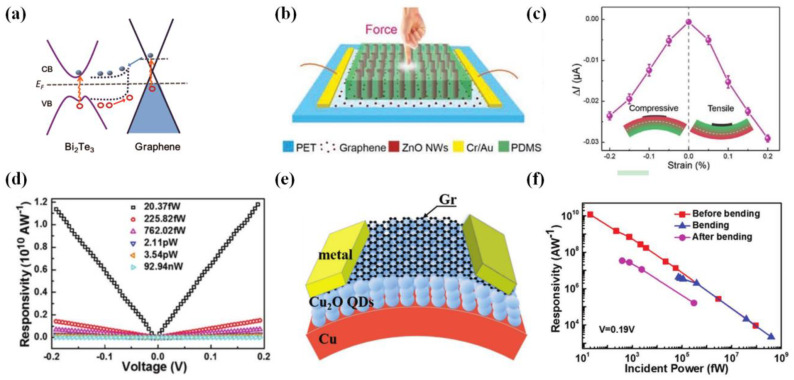
(**a**) Band diagram of a graphene Bi_2_Te_3_ heterojunction. Blue dots represent photogenerated electrons, while red hollow dots represent holes. Reproduced with permission [87]. Copyright © 2022, American Chemical Society. (**b**) Schematic diagram of the graphene/ZnO nanowire array device. (**c**) Current response for the different bending strains, where negative and positive strains denote compressive (left inset) and tensile (right inset) strains, respectively. Reproduced with permission [98]. Copyright © 2022 WILEY-VCH Verlag GmbH & Co. KGaA, Weinheim. (**d**) Responsivity as a function of the bias under different incident powers (under 450 nm illumination). (**e**) Illustration of the bent PD. (**f**) Responsivity versus incident power for the PD before bending, bending with a curvature of 40 mm, and after bending. Reproduced with permission [101]. Copyright © 2022 WILEY-VCH Verlag GmbH & Co. KGaA, Weinheim.

Cuprous oxide is a non-toxic, naturally p-doped material with a doping level of N_A_ < 10^15^ cm^−3^, which is obtained via the thermal oxidation of copper and exhibits a high light absorption coefficient, high electrical conductivity, and high hole mobility [102]. Liu et al. [101] used non-transferred graphene and Cu_2_O to prepare a graphene/Cu_2_O QD hybrid PD, which showed an ultrahigh responsivity of ≈10^10^ A/W at room temperature and achieved fW-scale optical detectivity. This device is flexible, with a responsivity that increases linearly with increasing bias, as shown in Figure 3d. Figure 3e shows that the PD can be bent with a curvature of 40 mm, and its responsivity is still approximately 4 × 10^6^ A/W at 72.3 pW. However, the device cannot detect weak light intensity at bending states. Even so, the close-to-inversion relationship between the responsivity and incident light intensity still holds, as shown in Figure 3f. The preparation of this device opens the pathway to realize the application of transfer-free graphene in highly sensitive optoelectronic devices. Compared with the complexity of the transfer-free manufacturing process of graphene, many more studies are still focused on transferring graphene to a substrate [103]. Monolayer graphene can be transferred to a Cu_2_O surface through carbon powder-assisted thermal oxidation. Thus, Cong et al. [104] prepared a graphene/Cu_2_O/Cu-based visible light PD. Under zero bias, this graphene/Cu_2_O/Cu PD exhibited an ultrafast response time of 4.1 μs at 530 nm, a responsivity at 550 nm of 86 mA/W, and high selectivity for visible light. It is important that the graphene/Cu_2_O/Cu PD can be bent to a radius of curvature of 30 mm, and even if it is bent thousands of times, it still maintains a high response rate.

Graphene-based composites composed of graphene and graphene QDs on stretchable substrates have been studied as sensitizing materials in hybrid structures. The combination of graphene and colloidal QDs with tunable light absorption also provides new possibilities for optoelectronic applications. This hybrid system can be used to address the issues associated with the number and mobility of photoelectric carriers [105]. The reported results showed that self-doped colloidal copper phosphide (Cu_3−*x*_P) NCs (nanocrystals) exhibited localized surface plasmon resonance and ultrafast carrier relaxation. Based on this, Sun et al. [106] produced a flexible PD composed of monolayer graphene and self-doped Cu_3−*x*_P QDs. This PD delivered a broadband light response in the range of 400–1550 nm. At a visible wavelength of 405 nm, the responsivity of the device is 1.59 × 10^5^ A/W, its photoconductor gain is as high as 6.66 × 10^5^, and it exhibits a high responsivity of 9.34 A/W at 1550 nm. The surface potential of the hybrid structure was studied, and it was found that the surface potential of the area covered by the NCs (doped graphene) was 175 meV higher than the one of undoped graphene, which reduced the work function of graphene. Comparative experiments proved that Cu_3−*x*_P NC surface ligands play a key role in determining the charge transfer efficiency of Cu_3−*x*_P to graphene. Through testing, the strain of the device is 1.95%, and after 5000 cycles of bending, the photocurrent and responsivity of the PD remain the same.

### 4.3. Graphene/Transition Metal Halide Heterostructure-Based PDs

Due to their flexibility and transparency, atomically thin vertical heterostructures have broad prospects in optoelectronic applications. The TMDs that are used to form these heterostructures are layered materials with strong in-plane and weak out-of-plane bonding. There are no dangling bonds on the surface of these materials. The atoms in the layers of the materials are connected via covalent bonds, and the layers are combined with van der Waals forces. They can be peeled off into a 2D layer with a thickness of a single cell. These materials have a unique layered structure and exhibit strong light–matter interactions, with high carrier mobility and a narrow bandgap [107,108]. The stacking of graphene and crystals of these 2D materials that exhibit extensive optical properties can combine the advantages of these two materials to prepare high-performance optoelectronic devices. Compared with traditional direct bandgap semiconductors, TMDs have excellent photoelectric properties and are easy to handle and process, showing excellent mechanical flexibility. Among the TMDs, tungsten disulfide (WS_2_) and molybdenum disulfide (MoS_2_) are the most researched materials for applications in optoelectronic devices in recent years. The reason for this is that their energy gap lies in the visible region of the electromagnetic spectrum, and they exhibit relatively high chemical stability [109]. MoS_2_ has excellent photoelectric properties, with a bandgap that can be turned from indirect to direct by adjusting the thickness of its layers. Photoelectric devices based on monolayerMoS_2_ nanosheets have a direct bandgap of 1.8 eV [110]. MoS_2_ can also be combined with graphene to form a Schottky barrier. Roy [111], Zhang [30] and coworkers combined graphene and 2D MoS_2_ to prepare photosensitive memory devices and photoconductive detectors, respectively. In these devices, the MoS_2_ layer is employed as the active layer to absorb photons and generate carriers. Under the application of a bias voltage, a light response of ~5 × 10^8^ A/W is achieved, which is currently the highest response rate exhibited for a graphene/TMD device. Similarly, limited by the trapped state inside the material and the defects at the interface, the response of the device is ~1 s. Recently, Zhou et al. [112] successfully realized the preparation of graphene/TMD-HB/graphene vertical devices, in which the top and bottom electrodes are both graphene and MoS_2_/WS_2_ is the active semiconductor layer in the middle, with a type II energy band formed between WS_2_ and MoS_2_. The photogenerated electrons (holes) are quickly separated and transferred from WS_2_ (MoS_2_) to MoS_2_ (WS_2_) under 532 nm light, thereby generating photovoltaic effects in the device. In addition, it is also found that the stacking symmetry determines that the type II directionality is the main factor controlling the direction of photocurrent under illumination, while the direction of photocurrent in a homogeneous double-layer film is not easy to control, as the doping level of graphene determines the tunnel barrier [113]. Due to the dielectric environment of graphene, the photocurrent seems to be fixed in the top and bottom graphene layers. Such vertically stacked layered heterostructures (VLHs) are usually not formed from a single 2D material, their quality of interface is high compared with those of traditional covalently bonded and thin-film heterostructures, and issues such as lattice mismatch strain due to a lack of covalent bonds will not be observed [112,114].

Subsequently, Kang et al. [115] produced a MoS_2_/graphene PD, which exhibited a high light responsivity of 6.3 A/W. As shown in Figure 4a–d, the graphene and MoS_2_ are transferred by a soft lithographic patterning process so that the two materials are stacked in a cross arrangement (Figure 4e). As shown in Figure 4f, the device is subjected to bending, and it is found that when the width of the MoS_2_ patterns is 10 μm, the photoresponse of the device has the least degradation upon bending. This manufacturing approach of 2D material cross-stacked patterns provides a new method for producing flexible, highly transparent PDs.

Recently, graphene/MoS_2_/graphene (GMG)-based high-performance PDs with horizontal heterostructures and vertical heterostructures have been extensively studied. The atoms in the layers of these heterogeneous structures are connected with covalent bonds, and the layers are combined through van der Waals forces. The core of the device is a Schottky junction formed between MoS_2_ and graphene, which shortens the transmission distance of photogenerated carriers between the source and drain electrodes and reduces the cutoff current. Compared with graphene- and MoS_2_-based biophotoelectric detectors, the response speed of GMG PDs is greatly accelerated. For example, the GMG lateral heterojunction PD synthesized by Liu et al. [116] using CVD exhibits high electrical characteristics, and the Raman spectrum is shown in Figure 4g. Electrical tests show that under different bias voltages, the switching current ratio is as high as 10^6^. The responsivity is more than 2 × 10^3^ mA/W, and the maximum *D** is as high as 10^13^ Jones. The test results show that in this GMG structure, the photocurrent increases with an increase in the illumination power and the grid voltage, and the transmission curves of the GMG PD under dark conditions and different illumination powers have almost the same characteristics (Figure 4h,i). This indicates that the GMG PD has an obvious gate-tunable photocurrent effect. Moreover, this lateral heterojunction functions as a high-performance sensor due to the unique electrostatic conditions on the interface [117]. Gao et al. [118] prepared a graphene/MoS_2_/graphene vertical heterostructure PD with excellent photoelectric detection. The structure is shown in Figure 4j. The working wavelength of this detector is 405–2000 nm, the responsivity at 532 nm is 414 A/W, the detectivity is 3.2 × 10^10^ Jones, the responsivity at 2000 nm is 376 A/W, and the detectivity is 2.9 × 10^10^ Jones. This device exhibited R with a greater value than that of a graphene/MoS_2_/graphene lateral heterostructure, which is 2 A/W. However, the manufacturing process of this vertical structure involves complex transfer and alignment, and its stability and controllability are not high, which has a certain impact on production efficiency.

**Figure 4 materials-15-04820-f004:**
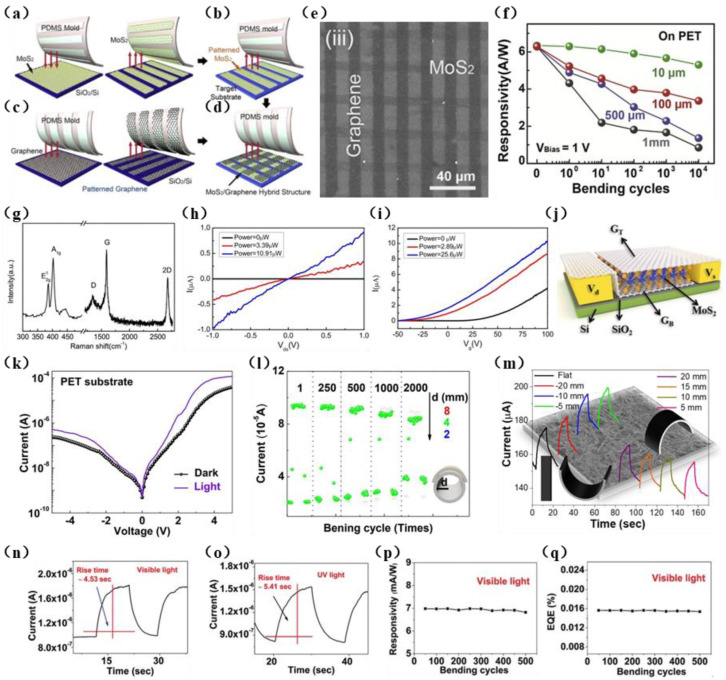
(**a**−**d**) Preparation process of MoS_2_/graphene PD. (**a**) CVD growth of MoS_2_ on a SiO_2_ substrate and transfer of the prepared MoS_2_ to the patterned PDMS stamp. (**b**) Transfer of MoS_2_ patterns on PDMS toward a target substrate. (**c**) Synthesis of the graphene layer via a thermal CVD method and transfer of it to a SiO_2_ substrate, patterning of the graphene, and its transfer to the PDMS stamp. (**d**) Transfer of the graphene lines to the target substrate to overlap the MoS_2_ layer vertically. (**e**) SEM image of the cross-stacked MoS_2_/graphene pattern. (**f**) The photoresponsivity of bending cycles (1−10,000) in PDs with different MoS_2_ pattern sizes at a bending radius of 9 mm and bias voltage of 1 V. Reproduced with permission [115]. Copyright © 2022 Elsevier Ltd. All rights reserved. (**g**) Raman spectra of graphene and MoS_2_, showing that graphene is a bilayer and MoS_2_ is a monolayer. (**h**) *I*_ds_-*V*_ds_ output characteristic curves. In the dark and under various illumination powers, V_g_ = 0. (**i**) Transfer curves for the GMG PD in the dark and under various illumination powers, V_ds_ = 1.0 V. Reproduced with permission [116]. Copyright © 2022 Published by Elsevier B.V. (**j**) GMG vertical heterostructure diagram. From bottom to top are graphene, MoS_2_, and graphene. Reproduced with permission [118]. Copyright © 2022 Wiley−VCH GmbH. (**k**) *I*-*V* curves of PD on a PET substrate in the dark as well as under illumination at *λ* = 532 nm. (**l**) Evolution of the repeated on/off current switching curves of the device over 2000 bending cycles (*d* = 8, 4, and 2 mm, *λ* = 532 nm). Reproduced with permission [119]. Copyright © 2022 Elsevier B.V. All rights reserved. (**m**) Response of a WS_2_/graphene heterostructure-based PD toward inward and outward bending. Reproduced with permission [120]. Copyright © 2022, American Chemical Society. (**n**,**o**) Rise time of SnS_2_/graphene under constant visible and UV irradiation. (**p**,**q**) Responsivity and EQE of the device under different bending cycles. Reproduced with permission [121]. Copyright ©2021, The Author(s).

Doping graphene with trifluoromethanesulfonyl amide (TFSA) and triethylenetetramine (TETA) molecules results in changes in its electronic composition. Unlike doping with metal particles, these two molecules tend to accept/donate electrons from/to the host materials, respectively, which induces p-/n-type doping [122]. Jung et al. [119] reported all-2D bisTFSA-doped graphene TFSA-graphene/MoS_2_/TETA-doped graphene vertical-heterostructure semitransparent PDs on rigid/flexible substrates. Figure 4k shows the *I*-*V* characteristics of the device in the dark and under illumination. The average visible light transmittance of this PD is 58%, and it has a responsivity of 0.128 A/W and a detectivity of 1.69 × 10^9^ cm Hz^1/2^/W when irradiated with 532 nm wavelength light. By adding an aluminum reflector to the semitransparent PD, *R* can be improved to 0.137 A/W, which is an increase of approximately 13%. Bending tests on this device showed that even after 2000 bending cycles (radius of curvature = 2 mm), the *R* value of the flexible PD was still maintained at approximately 32% (Figure 4l), and it exhibited excellent bending stability.

Among the various 2D layered metal disulfides, SnS_2_ has a high optical absorption coefficient in the visible region and excellent photoconductive properties. It is an n-type semiconductor with a layered cadmium iodide (CdI_2_) structure and a bandgap of approximately 2.2 eV [123]. Owing to its high chemical stability, low cost, unique structure, and environmental friendliness, SnS_2_ has received an extensive amount of attention, especially in photoelectric detection. Using a facile chemical bath deposition method (CBD) to prepare a vertically grown SnS_2_ nanosheet array film, Fang et al. [124] obtained a graphene/SnS_2_ nanosheet array thin-film heterojunction PD. Then, they adopted a direct transfer using a wet chemical method to introduce a transparent graphene film as a hole-transport layer in a PD. The composition and microstructure of the obtained samples were analyzed by Raman spectroscopy, and the results indicated that a graphene/SnS_2_ composite was successfully fabricated. The detector exhibits strong photoresponsivity, with a measured optical on/off ratio of 1.53, which is higher than that of pure SnS_2_ film of 1.1. The response time (rise time/decay time) was 5 s/4.9 s, which was better than that of pure SnS_2_ array membranes (5.4 s/5.1 s). The improvement in photoresponse indicated that the effective heterojunction formed between the vertical SnS_2_ nanosheet array film and graphene suppresses the recombination of photogenerated carriers [125]. This successful photoelectric enhancement opens up new opportunities for applications of other low-temperature soluble transition metal sulfides and graphene composites as PDs.

In 2020, Tweedie et al. [126] developed a graphene single-layer WS_2_-graphene lateral heterostructure PD on a PEN substrate, which failed during strain measurements. Later, Pataniya et al. [120] reported a low-cost and green synthesis approach using WS_2_ to decorate pencil trace on ordinary cellulose paper to prepare a high-performance WS_2_/graphene heterostructure PD. It is noteworthy that cellulose paper is readily available, light weight, and environmentally friendly. The production of such paper-based flexible equipment is not only a huge advantage in terms of cost, but also allows environmentally friendly and super large-area equipment to be prepared. The fabricated device uses large-area electrophoretic deposition technology to decorate the pencil trace conductive film with liquid-phase exfoliated WS_2_ nanosheets. The resultant device exhibited excellent photoresponse and temporal photoresponse characteristics over broad spectral regions, from the visible to near-IR region. This is due to the WS_2_ nanosheets having the ability to absorb photons over a broad spectral region and the high electrical conductivity of graphene promoting the unimpeded flow of photogenerated carriers [127]. After optimization of the device’s structure, the responsivity of the device is 0.439 A/W, the detection rate is 1.41 × 10^10^ Jones, the EQE is 81.39%, and the response time is 2.1 s. As shown in Figure 4m, the bending diameter of the PD changes from 5 mm to 20 mm. After preliminary observations, it is shown that the dark (baseline) current changes significantly upon bending of the detector. The response of the device was investigated over 500 bending cycles, and it was found that the PD showed a stable and reasonable photoresponse during cycling [120]. Additionally, Selamneni et al. [121] dip-coated biodegradable cellulose paper substrate with graphene and then directly grew SnS_2_ on the substrate via a hydrothermal method to prepare a SnS_2_/graphene broadband heterojunction PD. Using first principles calculations, the bandgap and electron affinity values are theoretically estimated. The test results showed that under visible and UV light, the response rates of the device were 6.98 and 3.67 mA/W, respectively, and the response times were 4.53 and 5.41 s, indicating that the response of the device to visible light was stronger than that to UV light (Figure 4n,o). The PD was tested over 500 bending cycles, and it was found that the change in its responsivity and EQE values was negligible, demonstrating the extremely strong flexibility of the device (Figure 4p,q). A large-area SnS_2_/graphene heterojunction was successfully fabricated on a low-cost cellulose paper substrate, and its performance metrics were found to be equivalent to those of equipment manufactured using sophisticated techniques, representing a major step in the development of low-cost PDs.

### 4.4. Halide Perovskite-Type Graphene Heterostructure PDs

Perovskite materials exhibit high light absorption characteristics. The photoluminescence of CsPbX3 nanocrystals is characterized by narrow emission line widths of 12–42 nm, wide color gamut covering up to 140% of the NTSC color standard, high quantum yields of up to 90%, and radiative lifetimes in the range of 1–29 ns [128]. In particular, halide-type perovskite materials have become a research hotspot for high-efficiency light-absorbing materials in optoelectronic devices in recent years due to their excellent optical properties [129,130]. Over the past decade, due to their outstanding physical properties, organic trihalide perovskites have attracted widespread attention in the field of optoelectronic applications. Dang et al. [131] developed a flexible perovskite/graphene hybrid channel field-effect phototransistor structure, as shown in Figure 5a. The main hole transfer from the perovskite to graphene and the 1.5 eV bandgap of the perovskite result in a phototransistor exhibiting a narrow and highly sensitive light response over the visible spectrum. These features result in the device exhibiting a maximum responsivity at 515 nm of 115 A/W, several times higher than those of flexible PDs based only on perovskites (3.9 A/W), and it also exhibited a detectivity of up to 3 × 10^12^ (515 nm). However, due to the long lifetime of photogenerated carriers, the response speed of the device increased (τ_rise_ ≈ 0.25 s, τ_decay_ ≈ 5.3 s). The PD is shown to withstand 3000 cycles of tensile strain with a radius of curvature of 12mm, with no obvious changes in photocurrent, indicating reliable light response stability after successful bending (Figure 5b,c). Under normal circumstances, the key for determining device performance lies in the grain boundaries and device structure of the perovskite film, especially for horizontally structured devices [132]. Zou et al. [133] produced the first optimized PD with a vertical structure, in which the single-crystalline perovskite MAPbBr_3_ was used as a light absorber and monolayer graphene as a charge transport layer. The hybrid device therefore combined the strong light absorption characteristics of the perovskite and the high carrier mobility of flexible graphene, exhibiting high light response performance under low-light intensity (0.66 mWcm^−2^) driven by a 3 V bias voltage, a high light response rate (≈1017.1 A/W), and high light detection performance (≈2.02 × 10^13^ Jones) under 532 nm laser irradiation. As shown in Figure 5d–g, the responsivity and detectivity of the device are functions of the incident light intensity due to the fast charge transfer in graphene and the long recombination life of the single-crystalline perovskite, resulting in an ultrahigh photoconductivity gain of ≈2.37 × 10^3^. The energy band diagram of the device under illumination is shown in Figure 5h.

In optoelectronic devices, the synergistic effects and complementarity of perovskite hybrid materials will result in improved performance of each component. However, the formation of spatially inefficient perovskite islands on graphene leads to the formation of perovskite grains with lower crystallinity, which reduces the photogenerated carrier density compared with the perovskite layer and remains an issue to be solved. A previously studied dual hybrid system did not exhibit a synergistic effect, which led to insufficient improvement in device performance [135]. Therefore, a problem to be solved in high-performance graphene hybrid PDs is to effectively combine high-efficiency plasma NPs with high-density perovskite films. Using gold nanostars (GNSs) and perovskite materials, Lee et al. [134] developed flexible high-performance graphene hybrid PDs exhibiting hot electron transfer and high-efficiency light trapping. As shown in Figure 5i, although the original graphene PDs do not show obvious photodetection characteristics due to their extremely low photon absorption and ultrafast charge carrier recombination, the graphene PDs functionalized with GNSs and densely covered perovskite layers exhibit excellent performance (Figure 5j), with the photocurrent level greatly increased in the graphene and perovskite hybrid system. Furthermore, due to the light trapping effects of the plasma GNSs, their introduction into the devices further increased their photocurrent. The *R* is 5.90 × 10^4^ A/W, with a D* of 1.31 × 10^13^ Jones, the highest detection value of a perovskite-functionalized graphene PD reported [136]. In addition, a flexible 10 × 10 PD array was produced. When the bending radius is less than 3 mm and the bending test is repeated 1000 times, the array exhibits high optical signal resolution spatiotemporal mapping and significant mechanical durability (Figure 5k–l). This perovskite-type plasma NP hybrid system can be easily applied over a large area and therefore is very suitable for making high-performance and flexible optical sensing systems.

Recently, Che et al. [137] demonstrated a CsPbBr_3_ NC/graphene hybrid PD that can be used to effectively detect UV light, in which CsPbBr_3_ NCs are used as a light absorber and graphene as a carrier transport channel. The hole and electron mobilities of the device in the dark are 1.7 × 10^3^ cm^2^/(V·s) and 6.4 × 10^3^ cm^2^/(V·s), respectively. Under the illumination of 405 nm light, the device exhibited high responsivity, D*, and EQE, with maximum values of 3.4 A/W, 7.5 × 10^8^ Jones, and 103%, respectively, and short rise/decay times of 7.9/125 ms, respectively. Although the response speed, responsivity, and stability of flexible graphene/perovskite hybrid PDs need to be further improved, this facile and low-cost manufacturing method provides the possibility for producing devices for large-scale applications in sensitive broadband detection [138]. These new flexible graphene hybrid PDs are expected to contribute to the development of imaging sensors suitable for low-light photography, wearable PDs, and UV detectors that exhibit high light responsiveness.

### 4.5. Other Graphene Hybrid PDs

The design of hybrid PDs aims to improve the sensitivity of graphene PDs, but the integration process of these hybrid systems is complicated. Combining graphene with QDs can also extend the lifetime of induced photon carriers, which helps to obtain devices that can achieve a high-response light detection distribution. Hybrid graphene–QDPDs combine the strong light absorption of QDs with the high mobility of graphene, in which the QDs absorb light and generate photogenerated carriers efficiently transported by graphene [139,140,141]. To enhance the efficiency and stability of graphene/Si QD (SQD)-embedded SiO_2_ (SQD:SiO_2_) ML heterojunction PDs, Shin et al. [142] employed bis (TFSA) as a dopant for graphene. The resultant optimized PD exhibited an *R* value of 0.413 A/W at a doping concentration of nD = 20 nm, a linear dynamic range of 92 dB, a detection efficiency of 1.09 × 10^10^ cmHz^1/2^/W, and an EQE of 81.33% at a peak wavelength of 630 nm. In air for 700 h, the device has almost no loss in the *R* value. These characteristics are comparable with those of commercially available Si-based PDs and are superior to those of previously reported graphene/Si PDs.

Graphene/SCs are another hybrid PD structure. In these structures, a heterojunction is formed between graphene and an SC. This heterojunction simultaneously has excellent light absorption of SCs and an ultrahigh migration rate of graphene, in which the photogenerated carriers are separated at the graphene/SC interface. This unique heterostructure breaks the limitations of previous epitaxial growth, such as point defects and dislocations caused by interatomic diffusion and lattice mismatch [143,144]. However, the D* of this graphene/SC phototransistor obtained from source-drain electrodes is assumed to be 1/*f* noise. Tian et al. [145] combined monolayer graphene and an n-GaAs epitaxial layer to prepare a graphene/GaAs heterojunction PD. The strong built-in electric field formed by the heterojunction region has a good spectral response. The D* measured by the source-gate electrode is approximately 1.82 × 10^11^ Jones. Compared with the results measured by the source-drain electrode, D* is increased by approximately 500 times. The reason for this result is that the Schottky barrier at the graphene/SC interface shields the process of carrier capture and release. In addition, the rise and fall times of the PD are 4 and 37 ms, respectively, and the response speed is increased by two orders of magnitude. By analyzing the rectification characteristics and photovoltaic characteristics of this type of PD, it is found that at 300 K, this type of PD is highly sensitive to visible/near-IR light (405–850 nm). These results further confirmed the great potential of the graphene/GaAs heterojunction as a high-performance broadband self-powered PD [144].

The current status and research progress of different graphene hybrid PDs are discussed. Through the combination of graphene and other materials, some traditional inorganic materials can be used in PDs, which exhibit excellent optical and electrical performance. The unique 2D structure of graphene makes an important contribution to the improvement of the detectivity and responsivity of these devices. Table 1 shows the parameters of different graphene-based hybrid PDs.

## 5. Application of PDs in Flexible Wearable Electronics

The design of flexible electronic devices needs to overcome the mechanical mismatch between rigid inorganic SCs and elastic and curved surfaces. There are two ways by which to build a structure that can bend: using flexible materials or making a flexible structure. When the material is thin enough, the bending strain decreases as the thickness of the material decreases. For example, single-layer or few-layer graphene atoms are thin and exhibit high flexibility. Graphene-based devices can be bent and cycled many times without affecting their performance by placing them on flexible substrates [12,155]. Recently, it was found that the special 3D structure of graphene heterojunction PDs promotes device functions that cannot be realized by planar geometric structures. The special properties of 2D materials ensure high mechanical flexibility of the resultant devices and allow front and simultaneous illumination on their back [156]. However, for advanced applications, such as integration with the human body, the simple bendability of a device is not enough. For such applications, the stretchability of the device needs to exceed 30% to match the elasticity of human skin. In 2016, a stretchable PD with enhanced strain tunable optical responsivity, in which graphene is folded on a stretchable substrate, increased the areal density of graphene through the development of this curved 3D structure. As a result, compared with the planar graphene PD, the absorbance of the device is increased, and the light response is increased by 370%. After testing its flexibility, it is found that the telescopic capacity of the PD is 200% of its original length, and its light response rate is improved by approximately 100%. That is, the flexible PD can be tuned mechanically, and upon an increase in its surface density, there is an increase in its absorption rate by 400% [157]. Figure 6a–d shows the highly retractable PD on the surface of a human brain model, where a consistent dynamic photoresponse is achieved with ≈11.1% induced tensile bending strain. In 2018, Ding and coworkers [158] prepared a retractable PD on a prestrained 3 M VHB substrate using perovskite CH_3_NH_3_PbI_3_ microwires and graphene as the matrix. Stretchability of the device is achieved by releasing and crumpling the perovskite/graphene matrix. The device performs well after being stretched to 50% strain 100 times. When stretched to 80% or even 100%, the photocurrent does not disappear completely, and the response is faster.

Although the manufacturing process of flexible stretchable devices is complicated, they have multiple functions. For example, a flexible PD can be placed on the surface of skin to form a wearable patch for the monitoring of human health. Choi et al. [159] developed a photoelectric device consisting of a curved image sensor (CurvIS) array and an ultrathin neural-interfacing electrode (UNE) by mimicking the structural characteristics of the human eye and using an anatomically thin MoS_2_-graphene heterostructure to be softly implanted into the retina of the human eye. MoS_2_ is transferred to the graphene electrode as a light-absorbing layer and can be used to effectively detect light signals and apply programmed electrical stimulation to the optic nerve, with minimal mechanical side effects on the retina, as shown in Figure 6e,f. The responsivity of this heterostructure device is two to three orders of magnitude higher than that of Si photodiodes of the same thickness, and it exhibits high imaging capabilities and great potential for use in biomedical applications.

As a very low-cost soft material, the light weight, flexibility, and environmental friendliness of paper have made it the focus of many studies [160]. Paper-based PDs make full use of the natural moisture absorption and flexibility of paper, exhibiting high flexibility. Recently, a new type of wearable paper-based PD was developed, which is the first PD that can be used to simultaneously measure the intensity and dose of UV light. Even after 1000 bends, the photoelectric signal of this PD remains stable. Part of the properties are shown in Figure 6g,h, with the highest responsivity at 360 nm, and the overall transparency of the material is also excellent [161]. Therefore, paper-based equipment exhibits excellent performance when used on curved surfaces, but the stretchability of the equipment is limited (<5%) [162,163]. Such highly deformable electronic devices represent a new direction for the development of future flexible electronics and can be applied in many areas, for example, in monitoring systems, optical imaging systems, optical communication, and image sensing.

**Figure 6 materials-15-04820-f006:**
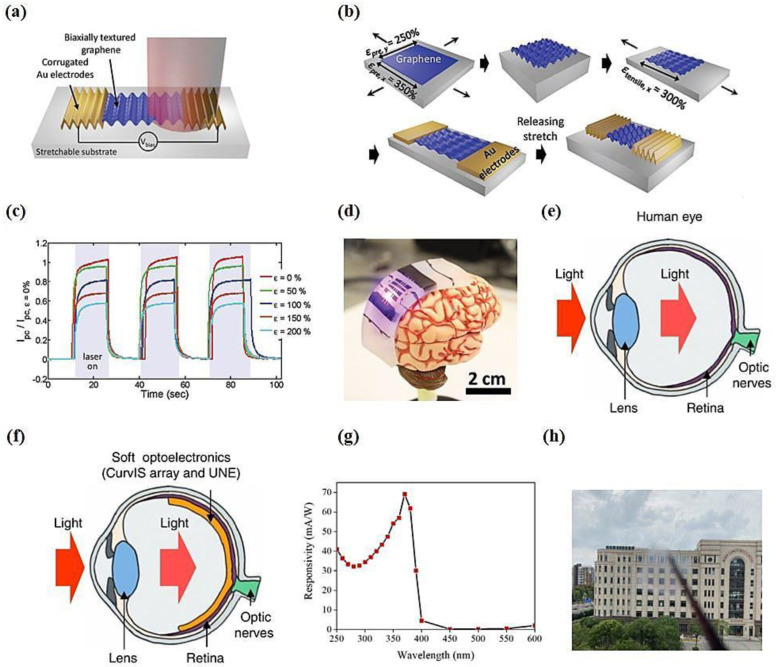
(**a**) Schematic diagram of a retractable PD and experimental device. (**b**) The manufacturing process of the stretchable graphene PD. (**c**) Dynamic photoresponse of the device at uniaxial tensile strains (ε_tensile,x_) varying from 0% to 200%. Measured photocurrent normalized to the photocurrent at ε_tensile,x_ = 0%. (**d**) Highly stretchable and conformal PD on the surface of a human brain model. Reproduced with permission [157]. Copyright © 2022 WILEY−VCH Verlag GmbH & Co. KGaA, Weinheim. (**e**) Schematic illustration showing the ocular structure of humans: the human eye consists of a lens that collects light, a retina that converts light into action potentials, and optic nerves that transmit action potentials to the brain. (**f**) Schematic illustration showing the ocular structure of the soft optoelectronic device of an ultrathin soft optoelectronic device consisting of a CurvIS array and a UNE conformally laminated on the hemispherical retina. Reproduced with permission [159]. Copyright © 2022, The Author(s). (**g**) The responsivity of the spray−coated ZnO nanoparticles under a 5 V bias. (**h**) Photo of the covering glass slide after Sr_2_Nb_3_O_10_ nanosheet spraying, showing the transparency of the material. Reproduced with permission [161]. Copyright © 2022 Wiley−VCH GmbH.

## 6. Conclusions and Outlook

The development of PDs has grown very rapidly over the last few years. Traditional graphene PDs based on composites have been extensively studied. With the ability to integrate them with soft materials and bendable surfaces, flexible electronic products surpass traditional rigid electronic device technology and have broad development prospects in wearable devices and the Internet of Things. This review discusses and summarizes the latest developments in graphene flexible PDs. Graphene can be used not only as a transparent conductive layer, but also as a light-absorbing layer. By combining graphene with different materials, the sensitivity of the resulting device can be significantly improved, and its detection spectrum can be extended from the UV to mid-IR or even THz bands. Graphene PDs can be applied to other types of sensing elements, such as gas sensors or biosensors, to achieve multifunctional sensing. Although great progress has been made in the photodetection of heterostructures, the charge transfer in heterojunctions still needs to be studied in depth. Although one of the performance indicators of a device can reach an ultrahigh level, the overall performance of PDs still needs to be improved. For applications in different fields, the focus on improving device performance is also different. For example, imaging applications require high responsiveness, while optical communication fields require devices that exhibit fast response. In practical applications, factors such as manufacturing costs and the size of packaging should also be taken into consideration. The quality of graphene presents a huge challenge in the large-scale production of graphene PDs. Although many methods can be used in the large-scale preparation of graphene films (as mentioned above), many challenges still exist. The preparation technology of graphene films is an issue that requires attention, in addition to the complexity and cost of the processes. Although the performances of graphene-based hybrid devices are far superior to those of traditional PDs, they are still not currently used in large-scale commercial applications. Research on new layered materials with suitable bandgaps and effective light absorption thus remains a critical challenge.

## Figures and Tables

**Figure 1 materials-15-04820-f001:**
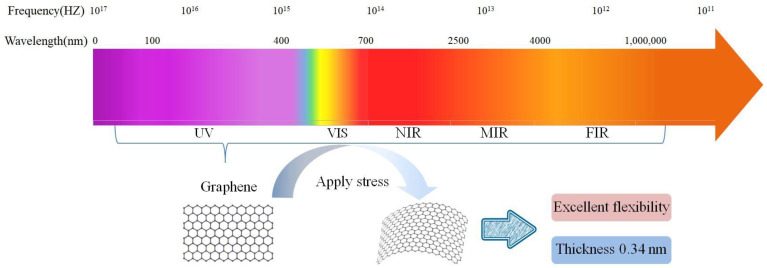
Bandgap values of various 2D materials and their corresponding detection ranges. Detection range and properties of two-dimensional material graphene.

**Figure 2 materials-15-04820-f002:**
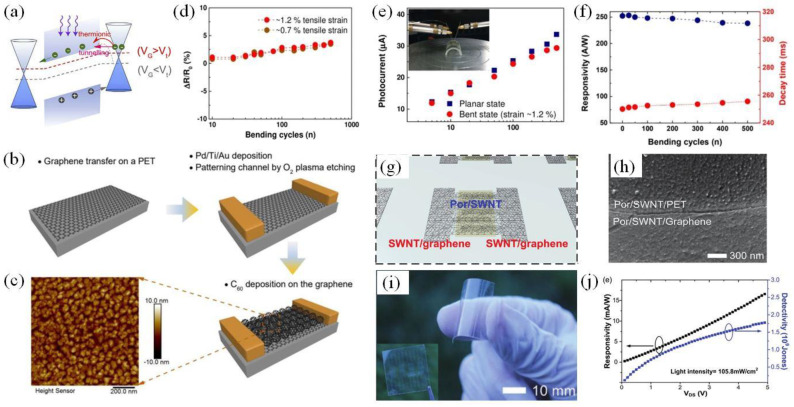
(**a**) Schematic of gate-tunable energy band diagrams of graphene−C_60_−graphene PD under illumination. Reproduced with permission [69]. Copyright © 2022 Elsevier Ltd. All rights reserved. (**b**) Schematic diagram of the manufacturing of graphene-C_60_ devices on plastic PET substrates. (**c**) Atomic force microscopy image of C_60_ on graphene film. (**d**) Relative change in resistance after 500 repeated bending cycles at 0.7% and 1.2% tensile strain. (**e**) Light intensity (laser) of 405 nm, photocurrent of the PD in planar and bent states. The inset shows an optical image of the device under tensile strain. (**f**) Changes in the responsivity (left axis, blue) and decay time (right axis, red) under ≈1.2% tensile strain after 500 bending cycles. Reproduced with permission [70]. Copyright © 2022 Elsevier Ltd. All rights reserved. (**g**) Graphene electrode, SWNT, and porphyrin layer fabricated on a plastic substrate. (**h**) SEM image of the boundary between the graphene electrode and the PET substrate after porphyrin functionalization. SWNTs covered by porphyrin molecules uniformly coated on graphene and PET substrates. (**i**) The prepared PD, exhibiting its high transparency and flexibility, where the lower left image shows the PD under normal conditions. (**j**) Optoelectronic characterization of a porphyrin-SWNT-graphene PD. Under a light intensity of 105.8 mW cm^−2^ and V_G_ of 0 V, the responsivity (left axis) and detectivity (right axis) as a function of V_DS_ reach maxima of ≈1.6 × 10^−2^ A/W and 2.6 × 10^9^ Jones, respectively, at V_DS_ = 5 V. Reproduced with permission [71]. Copyright © 2022 WILEY-VCH Verlag GmbH & Co. KGaA, Weinheim.

**Figure 5 materials-15-04820-f005:**
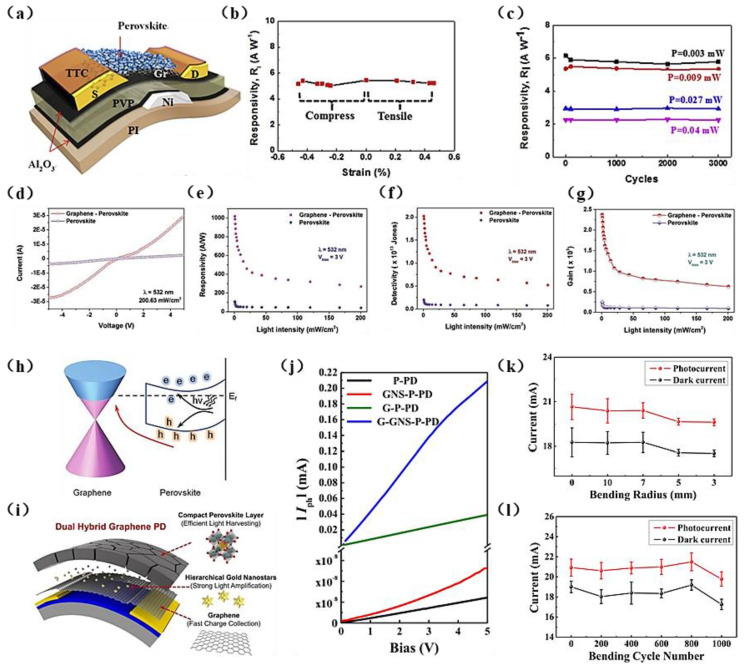
(**a**) Schematic diagram of a cross section of the flexible PD device. (**b**) Photocurrent responsivity (RI) of the flexible PD under various strain conditions, from a compressive strain of 0.46% to a tensile strain of 0.4% at a wavelength of 515 nm. (**c**) RI of the flexible PD prior to and after cyclic bending for up to 3000 cycles at a bending radius of 12 mm. Reproduced with permission [131]. Copyright © 2022 Elsevier Ltd. All rights reserved. (**d**) *I*−*V* characteristics of perovskite and graphene–perovskite vertical PDs under 532 nm illumination. (**e**–**g**) Responsivity, detectivity, and gain of perovskite and graphene–perovskite vertical PDs under 532 nm illumination at a 3 V bias. (**h**) Energy band diagram of a graphene–perovskite vertical device under illumination. Reproduced with permission [133]. Copyright © 2022 WILEY−VCH Verlag GmbH & Co. KGaA, Weinheim. (**i**) Schematic illustration of a graphene double-hybrid PD composed of a perovskite material and GNSs. (**j**) *I*_ph_−*V* characteristics of the P−PDs, GNS−P−PDs, G−P−PDs, and G−GNS−P−PDs under 10 μWcm^−2^ blue light (460 nm), (**k**) under various bending radii (down to 3 mm), and (**l**) during various bending cycles (up to 1000 cycles) under polychromatic light. Reproduced with permission [134]. Copyright © 2022, The Author(s).

**Table 1 materials-15-04820-t001:** Parameters of different PDs.

Materials	Wavelength (nm)	R (A/W)	D* (Jones)	Time	Ref.
**I Graphene-carbon (allotrope)**					
Graphene/SWNTs	532	51	—	40 ms	[78]
Graphene/C_60_	405	10^4^	—	76 µs	[70]
**II Graphene chemical doping**					
Graphene/Bi_2_Te_3_	633	20.5	—	210 µs	[146]
Bi_2_Te_3_ (NWs)/graphene	2200	0.09 ma/W	—	—	[17]
ZnO NRs/graphene	365	2.5 × 10^6^	—	—	[96]
Graphene/Cu_2_O/Cu	550	86 ma/W	—	4.1 µs	[104]
**III Graphene-TMDs**					
Graphene/MoS_2_/graphenelateral heterostructure	532	2 × 10^3^ ma/W	10^13^	1.8 s	[116]
Graphene/MoS_2_/graphene vertical heterostructure	2000	376	2.9 × 10^10^	—	[118]
TFSA-GR/MoS_2_/TETA-GR	532	0.128	1.69 × 10^9^	—	[119]
WS_2_/graphene	390~1080	0.439	1.41 × 10^10^	2.1 s	[120]
SnS_2_/graphene	380~780	6.98 ma/W	—	4.53 s	[121]
**IV Graphene-halide perovskites**					
Graphene/MAPbBr_3_	532	1017.1	2.02 × 10^13^	—	[133]
CsPbX_3_ NCs/graphene	405	3.4	7.5 × 10^8^	7.9/125 ms	[137]
FAPbI_3_perovskite/graphene	515	115	3 × 10^12^	0.25/5.3 s	[131]
Perovskite/GNs/graphene	532	5.9 × 10^4^	1.31 × 10^13^	—	[134]
**V Mixed structures**					
Graphene/ZnO NWs	365	3.2 × 10^4^	—	—	[147]
Graphene/MoS_2_	400–1000	10	—	0.28/1.5 s	[148]
Graphene/WSe_2_	532	350	1 × 10^13^	50/30 µs	[149]
Graphene/TiO_2_/p-Si	750	3.6	4 × 10^13^	—	[150]
Graphene/PdSe_2_	650–1550	6.68 × 10^4^	—	660 µs	[151]
Graphene/PSB QDS	808	4.2 × 10^2^	2.1 × 10^9^	10.6 ms	[152]
Graphene/TiO_2_	532	179	9.12 × 10^9^	20 ms	[153]
Graphene/WSe_2_nanosheets	670	6.66	1.94 × 10^8^	0.8/1.4 s	[154]

## Data Availability

Not applicable.

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
