# Peer review of "Recent Progress on Graphene Flexible Photodetectors"

_materials, 2022, doi:10.3390/ma15144820_

Round 1
Reviewer 1 Report
Well done.
Author Response
Thank you for your careful review. We wish good health to you, your family, and community.
Reviewer 2 Report
Manuscript: 1748206
The review article “Recent progress on graphene flexible photodetectors” by M. Wang and colleagues present research progress in graphene-based photodetectors. The review covers important topics related to flexible photodetectors based on graphene, and other hybrid configurations, the fabrication processes, and include recent examples from the literature. Although the material covers most of the background in PDs there is substantial quantitative and comparative information missing that makes the comparison of these type of PDs with similar or traditional semiconductors challenging. I strongly recommend the authors to address my comments below.
Major concerns:
- In effective reviews the incorporation of key metrics and figures of merits is essential to perform quantitative comparisons between fabrication methods, device performances, etc. This review does this at some extent, especially in the last sections, but there is still considerable points to address. Some examples, but not the only ones, are:
Line 158: “In 2009, the Ruoff group successfully prepared a large-area graphene film on Cu foil for the first time via CVD.” How much large area is, or compared to what.
Line 405: “The light-graphene interaction and optical absorption are greatly enhanced, and the light responsiveness of the device is also greatly improved”. By how much?
Line 412: “Bismuth telluride (Bi2Te3), as a typical topological insulator, is similar to graphene, exhibiting hexagonal symmetry, a narrow bandgap, high carrier mobility, high stability, and great prospects in PDs.” Higher carrier mobility by how much?
Line 701: “Perovskite materials exhibit high light absorption characteristics” How much is high, or compare to what.
- One of the most important parameters for PDs is the wavelength of detection. As stated in the manuscript, graphene has a huge bandwidth for detection which. However, what dictates the bandwidth in the graphene-based PDs is not explicit in the manuscript.
- In addition, the low absorption coefficient posses engineering challenges in practical devices. Authors miss the opportunity to discuss options for responsivity enhancement. In line 303 “Among them, plasma enhancement, resonant cavity enhancement, and heterostructure recombination are widely explored.” authors include few options for enhancement, but not fine details are include, nor reference to those enhancement mechanisms are included.
Minor concerns:
- The content in figure 1 does not reflect the suggested information written in the caption.
- Some acronyms were not defined. Line 122, TCFs is not defined. Line 330, SCPDs not defined.
- Line 281: “These results are several orders of magnitude higher than previously reported results.” Which previously reported results does this statement refers to?
- One of the main advantages of graphene as the active material for the PDs is the zero band gap, but at the same time it could add challenges in functional devices. For example, if the PD is designed to detect visible light, what are the intrinsic or extrinsic mechanism to filter out other wavelengths, such as IR or UV for example? The authors do not cover or mention this challenge.
-These examples should be cited individually. Line 303 “Among them, plasma enhancement, resonant cavity enhancement, and heterostructure recombination are widely explored.”
- Line 295: The used of QDs layer to increase the photorespondivity in PDs is unclear. More information, descriptions will aid in conveying better the concept to the audience.
- Line 519: “It is important that the graphene/Cu2O/Cu PD can be bent to a radius of curvature of 30 nm, and even if it is bent thousands of times, it still maintains a high response rate.” Bending to 30 nm radius of curvature seems extremely low, is this a typo?
- The response and decay times in hybrid photodetectors of in the range of 5s/4.9s seems too large. See line 661. Although this is a property of the material/system, can the authors include what is the fundamental aspect the dictates such a large response and decay times?
Author Response
Thank you for your careful review. We wish good health to you, your family, and community. Our reply has been uploaded to the attachment, please check it.

Author Response

(The authors gave the same response as above.)

Round 2
Reviewer 2 Report
I appreciate the effort the authors made to address my and other reviewers' comments. I recommend this manuscript for publication in the present form.